# Response of Winter Wheat (*Triticum aestivum* L.) Yield to the Increasing Weather Fluctuations in a Continental Region of Four-Season Climate

László Huzsvai [1] , József Zsembeli [2,*] , Elza Kovács [3] and Csaba Juhász [3]

1    Faculty of Economics and Business, Institute of Statistics and Methodology, University of Debrecen, Böszörményi 138, H-4032 Debrecen, Hungary; huzsvai.laszlo@econ.unideb.hu
2    Research Institute of Karcag, Hungarian University of Agriculture and Life Sciences, Kisújszállási 166, H-5300 Karcag, Hungary
3    Institute of Land Use, Engineering and Precision Farming Technology, Faculty of Agricultural and Food Sciences and Environmental Management, University of Debrecen, Böszörményi 138, H-4032 Debrecen, Hungary; ekovacs@agr.unideb.hu (E.K.); juhasz@agr.unideb.hu (C.J.)
*    Correspondence: zsembeli.jozsef@uni-mate.hu; Tel.: +36-59-500-360

**Abstract:** Wheat is grown in the largest area in the world as well as in Hungary. Globally, the yield is predicted to decrease due to climate change; however, technological development can potentially compensate for it. In this study, the contribution of climatic and technological trends to the change in winter wheat yield in four sub-regions of Hungary with considerable spatial and temporal variations in weather conditions was evaluated. Long-term trends in both the weather conditions and the technology development, with the consideration of the socio-economic circumstances, were identified. For future yield prediction, non-climatic influences and critical climatic factors, as well as sensitivity in the phenological stages, were considered. In the past 50 years, the average yield variation was lower at regional than country scale. Winter wheat yield was not found to be sensitive to temperature, global degree days, precipitation, and climatic water balance, only to heat stress. Considering the technological development and the heat stress during the critical weeks in the last 30 years, an increase of yields can be expected by 2050 in Hungary's western regions (0.72–1.55 t ha$^{-1}$), while yield depression is predicted (0.27–0.75 t ha$^{-1}$) in the eastern regions compared to the values estimated for 2019, $\pm$1.5 t ha$^{-1}$ within a 95% confidence interval. We proved that yield estimations can show contradictory changes by sub-regions of an agricultural region if the contribution of site-specific technology development, the dominant weather stressor, and the most sensitive phenological phase is involved in the statistical analyses. Identification of the dominant climatic stressor(s) for the different crops is necessary to keep high yield or even increase it under the changing environmental circumstances. Our findings suggest that heat stress is the main concern to maximize winter wheat production in temperate climate zones.

**Keywords:** climate change; heat stress; yield prediction; site-specific evaluation



## 1. Introduction

Globally, wheat is the second most-produced cereal grain, after maize. In relation to the consequences of climate change, a 6.0 $\pm$ 2.9% loss in global wheat yield with each degree Celsius increase was predicted [1]. Simulated long-term average yields between 1981 and 2010 varied widely across 30 global locations. Values decreased by 1–28% with an increase of 2 °C and 6–55% in the case of a 4 °C temperature increase [2]. Already two decades ago, it was highlighted that an increase in extreme weather events in Europe would cause lower harvestable yields and higher yield variability [3]. For locations at lower latitudes, the increase in the simulated yield variability with a higher temperature was found to be more marked, and the relative yield decline was greater. The simulated potential yield of

wheat, calculated for the period of 1976–2005, showed a slightly positive trend in the United Kingdom, Germany, Belgium, and the Netherlands (0.04–0.06 t/ha/year), while in Italy and in the Central Eastern European countries, the trends were found to be negative in the range of −0.04 to −0.09 t/ha/year. In most regions, no trends were identified [4]. In Hungary, in Central Eastern Europe, the exposure of wheat production to climatic conditions was proved to exceed those detected at global scale [5].

It must be noted that winter and spring wheat yields are differently impacted by the changes in climatic conditions. Winter wheat (*Triticum aestivum* L.) sown in autumn requires exposure to low temperature, but without severe frost in the growth stage, to trigger the reproductive stage in spring. Spring wheat is grown in regions where winter temperature is extremely low. For example, in regions of Russia, stronger negative association between the yield and the heat stress for spring wheat was found, while the reverse was found for winter wheat [6]. Correlation between the atmospheric blocking duration and yield was found significant only for spring wheat. In terms of the probability of increasing frequency of heat stress days around flowering, winter wheat was less sensitive to the increasing $CO_2$ eq [7].

Based on long-term historical data, in the North China Plain, winter wheat yield improvement using cultivars was reported at 24.7% in the 1990s and at 52.0% in the 2020s, compared to results reported from the 1980s [8]. Contributions of soil fertility and chemical fertilizer input were calculated at 7.4% and 6.8% in the two periods, respectively, while the seasonal yield variation was found to be caused by weather factors around −39% to 20%. The contributions of cultivars, fertilization management, and climate change to the yield improvement in the period of 1980–2009 were 12.2–22.6%, 2.1–3.6%, and (−3.0)–3.0% for the same region [9]. In Australia, climate trends were found to be responsible for 30–50% of the observed increase in wheat yields between 1952 and 1996, with increases in minimum temperatures identified as having the dominant influence [10].

In the case of winter wheat, drought and heat stress are the two most widely investigated climatic stressors. Drought affects both leaf expansion and photosynthetic performance. Differences in biomass accumulation were found to result from the biomass accumulation from mid-flowering to the final yield, dominantly [11]. The duration and the rate of the grain filling stage determine the final grain weight. High temperature and drought are the major stressors during the maturation and ripening of cereals. Water limitation and high temperature during the grain development lead to considerable yield losses mainly caused by the reduction in starch accumulation [12]. The nitrogen–water limited conditions were reported to result in negative yield changes in the case of winter crops compared to water limitation alone [13]. For selected cultivars in Central Europe, in the period of 1991–2014, yield decreases resulting from a temperature of ≥31 °C and ≥35 °C around heading of approximately 10–18% and 10–22% were reported, respectively, and a 14–23% decrease resulting from drought after sowing. Cultivars performed well under drought conditions from heading to maturity, showing a yield increase of 10%. Heavy rainfall resulted in an approximately 10–20% decrease, while frost of −15 °C was reported to decrease the yield by 12–18% [14]. Considering the response of winter wheat in the case of heat stress, the individual grain yield weight decreased linearly with an increasing duration of heat stress in the range of 2 to 30 days when imposed at the start of grain filling, in a pot experiment [15]. In another report on the heat stability of selected winter wheat cultivars, heat stress of 42 °C applied 7 days after anthesis and kept for 2 days reduced the yield by 20.8–28.6% [16]. Based on a literature review conducted in 1999 for wheat, $T_{min}$, $T_{opt}$, and $T_{max}$ of anthesis are 9.5 °C, 21.0 °C, and 31.0 °C, respectively, while in the period of grain filling, these are 9.2 °C, 20.7 °C, and 35.4 °C, respectively [17]. In the period of 1972–2013 across western Germany, a negative linear trend for the heading day, the day of emergence, and the length of the vegetative phase of winter wheat was found [18]. It is not possible to define a general relationship between the temperature and the rate of development in the different development phases and for all wheat varieties. Stress factors such as heat or drought can be biased by the effects of cultivar changes on crop



phenology [17,18]. Directly linking temperature increase to changes in phenology results in overestimation of the sensitivity. Plants exhibit different morphological, physiological, biochemical, and molecular alterations resulting in stress tolerance under heat stress [19,20] or combined heat and drought stress [21]. Different approaches, such as drought severity indicators and ecophysiological crop models used for predictions for the same conditions, can give large differences in the estimated yield losses and trends over time [22].

Predictions are made mostly by using simulation models with the consideration of the plant–soil–atmosphere interactions. For the period of 2021–2050 in China, the growing season average temperature and precipitation were predicted to increase the winter wheat yield by 1.47% and 2.16%, with the scenario of 650 and 1370 ppm $CO_2$-equivalent, respectively [23]. In India, reduced wheat yield in areas with mean seasonal maximum and minimum temperatures was projected in excess of 27 °C and 13 °C, respectively, despite $CO_2$ fertilization benefits, by 2050 [24]. For cereal-growing areas of Great Britain, climate change is expected to negatively impact the level of evapotranspiration, and the moderate winter wheat yield gain caused by $CO_2$ fertilization may be lost after 2050 [25]. For Germany, representing the continental region of Europe, using four different crop models, a 9% yield increase was predicted under rain-fed conditions by 2050, if 540 ppm $CO_2$ concentration is reached [26]. For Italy, in the Mediterranean area, considerable change was not predicted, based on the weather trends between 1982 and 2012. Based on experiments, both positive and negative interactions of temperature and increased [$CO_2$] on yield were found. Yield was reduced by warmer mean seasonal temperatures and increased by doubling [$CO_2$], but the effect varied greatly between years and with temperature (7–168%) [27]. A steady increase in atmospheric $CO_2$ concentration likely has a positive effect on yield but is counterbalanced by increases in temperature. In harsh environments, its effect can be low or even nil [28]. It is site-specific that the effect of increased $CO_2$ is sufficient to balance the negative effect of water shortage on crop growth; based on a scenario analysis for the productivity, shoot N assimilation and its inhibition under increased $CO_2$ were suggested to be incorporated into the crop models that describe downregulation of photosynthetic processes [29]. In Denmark, for the period of 1985–2040, an 8.0% yield decrease was predicted when agroclimatic and soil conditions were considered [30]. Across Europe, within the period of 2010–2030, climate change scenarios show a shorter duration from sowing to end-of-grain filling by 2 weeks to 2 months [31]. From 2014 to 2060, the sowing date of winter wheat was calculated to be later by 15 ± 7 days and the maturity date to be earlier by two weeks on average, linked to enhanced crop development rate and higher simulated temperature during sowing to maturity [32]. Anthesis and maturity were predicted to be earlier by 13.2 days and 17.5 days across Europe, respectively, for the scenario of 2055 compared with 1960–1990 [33].

There are numerous low-scale spatial and temporal yield projections based on trend analyses of historical datasets; however, the outputs are often contradictory, even if the inputs are rather similar. In the case of statistical models, the determinative weather conditions and the critical phenological phases are not taken into consideration [4]. Earlier projections are sometimes found to be invalid later [34]. Even in the short term, differences were found in the spatiotemporal patterns of vulnerability in crop production and climate, for example, between 1995 and 2014 across Northwest China, with the change from high to low with geographic extent moving from southeast to northwest [35]. Winter wheat yields are susceptible to changes in many aspects of climate that do not overlap by region [36]. There are a few studies revealing opposite trends in the projected wheat grain yields for the same climate zone or agricultural area, published quite recently, e.g., for Northern China with continental climate [37], for the Guanzhong Plain of Northwest China [38], and for the UK [39]. In this respect, however, there is a shortage of knowledge for European temperate climate. Hungary is ranked sixth globally on the list of arable land per country area, with 48.5% (recorded in 2018), and produced 5.38 million metric tons of wheat on an area of 1.016 million ha in 2019, ranking Hungary 24th globally [40]. Despite its small area, weather conditions can be considered relatively highly variable.

At country scale, economic optimization of crop production, tailored to the projected environmental, such as climate change, and non-climatic factors, such as economic changes, should be based on a spatial and temporal assessment of higher resolution. The inter-annual variability in yields is controlled by different non-climatic and climatic factors that can be revealed only by site-specific statistical analysis. In general, technological development is not considered in the simulation models [33], while in the case of the statistical models, the determinative weather conditions and critical phenological phases are not pre-selected [4]. These are sources of over- or underestimations.

In this study, the contribution of both technological and climate trends to change in winter wheat yields was evaluated for four selected sub-regions located in Hungary. For future yield prediction, non-climatic influences and critical climatic factors were identified, and sensitivity in the phenological stages was considered. Based on our previous study conducted for maize [41], we assumed that in the investigated area, winter wheat grain yield is sensitive to changing sub-regional conditions in the long term, but the changes in the projected yields can show contradictory trends. Our objectives were the following: (1) to describe the trends in winter wheat grain yields in Hungary in the last 50 years; (2) to describe the climatic conditions in the vegetation period of winter wheat by sub-region; (3) to identify the critical weather variables in balanced winter wheat production; and (4) to give a prognosis for the regional winter wheat grain yields in 2050, considering both the climatic and non-climatic, i.e., technological development trends found to be affecting variables, based on long-term time-series datasets.

## 2. Materials and Methods

### 2.1. Study Area and Meteorological and Crop Yield Databases

According to the climatic classification of the Intergovernmental Panel on Climate Change, cold temperate dry (CTED) and warm temperate dry (WTED) climate zones are distinguished for Hungary [42]. In the CTED zone, the mean annual temperature is below 10 °C, while in the WTED zone, it is above 10 °C. The annual precipitation is less than the evapotranspiration in both zones. The locations of four selected regions with the relevant regional meteorological data assessed in this study were the same as given and described in our previous study on the performance of maize [41]. The Southeast region belongs to the category of WTED, the majority of the area of the Western region is covered by the CTED category, while the Southwest and the Eastern regions mostly fall into the WTED category, with some CTED (Figure 1).

For the assessment, the meteorological database of the National Meteorological Service of Hungary was used. Daily data of a 50-year-long time series were analyzed for minimum temperature ($T_{min}$), maximum temperature ($T_{max}$), precipitation, global radiation, and the number of sunshine hours. The wind velocity and relative air humidity data were estimated from the monthly recorded data. In Hungary, winter wheat was not irrigated in the investigated period of time.

Yearly yield data of winter wheat (*Triticum aestivum* L.) for the regions in the period of 1970–2019 were provided by the Hungarian Central Statistical Office [43]. The databases were homogenized and validated with no data gaps. The source of the spatial data was the Research Institute of Agricultural Economics of the National Agricultural and Innovation Center [44].

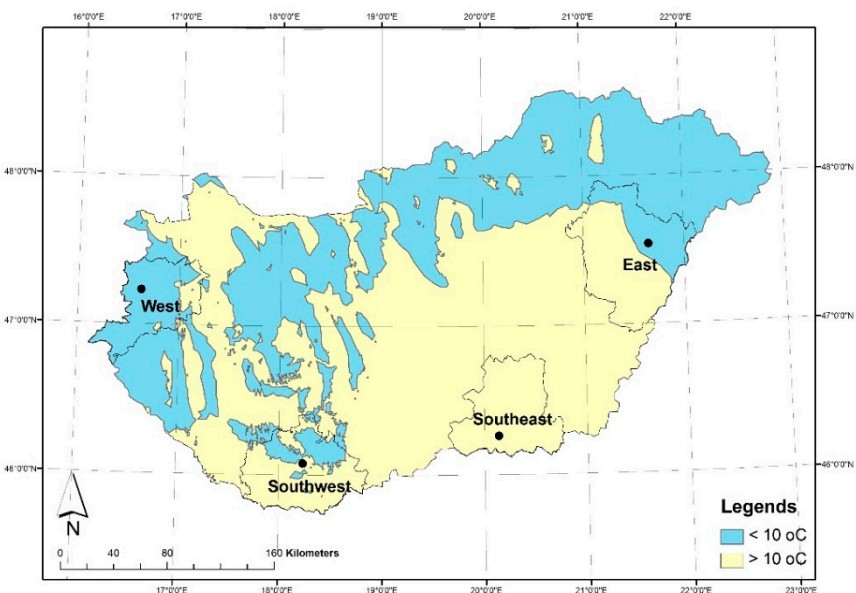

**Figure 1.** Locations of the four investigated Hungarian regions (West, Southwest, Southeast, and East) by climate zones according to the climatic classification of the Intergovernmental Panel on Climate Change. Blue color represents the areas where mean annual temperature is below 10 °C, while yellow where it is above 10 °C.

### 2.2. Data Analyses

Seasonal meteorological parameters, such as growing degree days (GDD, °C days), potential evapotranspiration ($ET_0$, mm), climatic water balance (CWB, mm), and heat stress units (HSU, °C), characteristic of the vegetation period (November–June) of wheat were calculated.

GDD values were calculated by using Equation (1). The mean daily air temperature was given by averaging the daily maximum and minimum air temperatures measured in 24 h period. If the mean temperature was below the base temperature ($T_{base}$), the GDD value for that day was zero [45] (*Method 1*).

$$GDD = \frac{T_{max} + T_{min}}{2} - T_{base} \tag{1}$$

The $T_{base}$ was considered 5.5 °C [46,47]. The upper threshold temperature was considered 25 °C [48]. If ($T_{max} + T_{min}$)/2 was above the upper threshold temperature, it was considered 25 °C. Growing degrees were accumulated on a daily basis over the vegetation period. The effect of photoperiod was not taken into consideration since the length of a calendar day at the same latitude and longitude is the same, and the planting time was considered to be the same.

The daily $ET_0$ values were determined according to the FAO-56 Penman–Monteith equation [48] and summed for the whole vegetation period of wheat. CWB was given as the difference between the sum of precipitation and the $ET_0$ in the vegetation period. The HSU was derived from the heat-sum of the daily $T_{max}$ values above 25 °C.

In the case of the mean air temperature, the total amount of precipitation, and the heat stress units, the weekly aggregates of the daily values were considered the independent variables in the modeling. For the GDD, $ET_0$, and CWB, data characteristic for the vegetation period were used for the analyses.

Critical weeks were considered by using Pearson's product-moment correlation. A week was considered critical when the absolute value of *r* was higher than 0.33 meaning the determination coefficient was around 10%. When heat stress occurred, the *r* was negative.

The main spring and summer phenophases of winter wheat in Hungary are summarized in Table 1.

**Table 1.** Main phenophases of winter wheat and their periods by date and number of calendar week.

|  | Period | No. of Calendar Week |
|---|---|---|
| stem elongation | 5 February–5 April | 5th–14th |
| heading | end of May, 4–9 days | 21st–22nd |
| anthesis | beginning of June, 4–6 days | 22nd |
| grain milk stage | 10–20 June, 10 days | 23rd–24th |
| grain dough stage | 20 June–1st July | 25th–26th |
| ripening | 1–20 July | 26th–29th |

Non-climatic influences, such as new cultivars and changes in crop management practices, were removed by detrending the yield. Weather parameters were also detrended (Equation (2)).

$$Y[t] = T[t] + S[t] + e[t] \tag{2}$$

where $Y[t]$ is the time-series value at period $t$, $T$ is the trend, $S$ is the seasonality, and $e$ is the residual component [49]. The trend of the yields was calculated by using linear regression analysis. The slope of the trend line was considered to describe the yearly technological development given in tons $ha^{-1}$ $year^{-1}$.

Normality was tested with the Shapiro–Wilk normality test ($\alpha = 0.05$). The detrended yields as the dependent variables were used for the multiple regression analysis. All explanatory climatic variables in the first model were included. A robust version of the Akaike model selection procedure was used for the regression model [50]. The significant variables were selected by using the stepwise regression method with backward elimination according to the Akaike information criterion [51]. For the future yield predictions, the trend in yield characterizing the technological development and the contribution of the climatic parameters were used.

Statistical computing was carried out using the R software [52], using version R 4.0.2 (22 June 2020).

## 3. Results and Discussion

### 3.1. Long-Term Winter Wheat Production in Hungary between 1970 and 2019

Winter wheat is a determinant crop in Hungarian crop production, with traditionally the largest production area. During the last 50 years, this area fluctuated within the range of 700 thousand to 1.4 million ha. During the period of 1970–1990, the production area of winter wheat was above 1.2 million ha, reaching or even exceeding 1.3 million ha in some years. From the 1990s, 1.1 to 1.2 million ha was characteristic, while during the last two decades, the production area declined down to 1–1.1 million ha. Taking the entire investigation period (1970–2019) into account, a decreasing tendency was observed.

The average yield of winter wheat shows an increasing tendency from the 1970s lasting until 1990 when the decade started with yields stagnating around 3.5–4 t $ha^{-1}$ (Figure 2). Another increase started only in the 2010s. The first increasing period in the 1970s and 1980s was due to the intensive development not only in crop production but also in the entirety of Hungarian agriculture [53]. This development was brought by the higher level of technical background and human expertise, the application of chemicals (e.g., fertilizers, pesticides, herbicides) [54,55]. Winter wheat production, as a considerable item in the national economy, reached its peak at the level of 6 million metric tons in the 1980s. Privatization of the land that started in 1989 had a negative effect on crop production, as discussed in our previous study [41], with winter wheat yields and production stability showing a significant decline. Recently, unfavorable conditions are partly compensated by the use of up-to-date varieties and hybrids that can be grown with high yield potential and good resistance against biotic and abiotic stresses. During the last decade, the least amount of winter wheat (3.7 million metric tons) was harvested in 2010. The peak was in 2016,

with 5.6 million metric tons. In a previous study, the estimated average of climate-driven yield loss based on actual and potential yields was reported to be higher in the period of 1951–1980 than that in 1981–2010, by 9.0% and 6.1%, respectively [5]. Here, we note that the authors did not consider the consequences of the change in the political regime in relation to the performance of the agricultural sector.

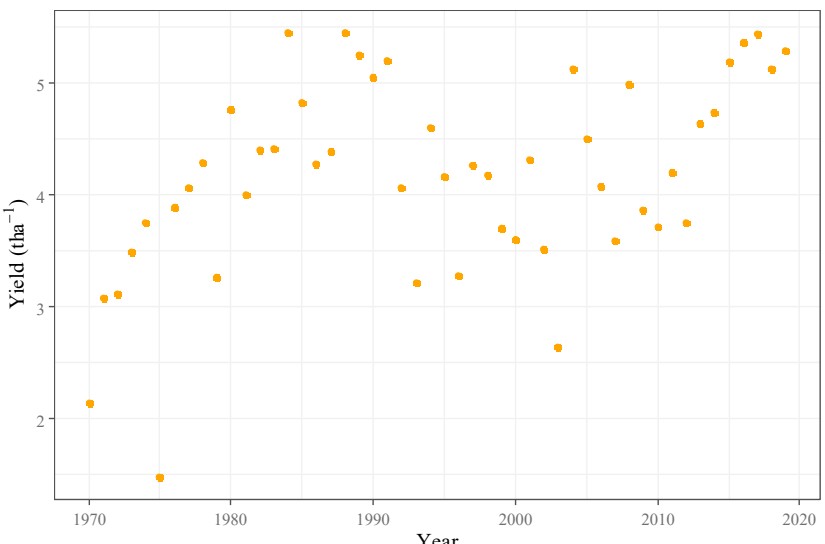

**Figure 2.** Average yields of winter wheat in Hungary (1970–2019). Source: Hungarian Central Statistical Office.

The yield of winter wheat in the periods of 1970–1989 and 1990–2019 is split in all four investigated regions according to the social and economic changes characteristic of Hungary. Interestingly, a bi-linear model with similar trends describes the evolution of grain yield in the four major producing countries of the European region—France, Germany, the United Kingdom, and Poland—in the last 80 years [56].

In both periods, yield data were normally distributed as the p-values were higher than 0.05 for all the regions (Table 2). Based on the results, it can be established that yield data were suitable for parametric statistical analysis.

**Table 2.** Results of the Shapiro–Wilk test proving the normal distribution of winter wheat yield data of the four investigated regions, in two time periods. W is the Shapiro–Wilk test statistic; *p* is the probability of error type I.

| | 1970–1989 | | 1990–2019 | |
|---|---|---|---|---|
| **Region** | **W** | ***p*-Value** | **W** | ***p*-Value** |
| 1. West | 0.974 | 0.827 | 0.977 | 0.728 |
| 2. East | 0.941 | 0.250 | 0.948 | 0.154 |
| 3. Southwest | 0.916 | 0.083 | 0.979 | 0.794 |
| 4. Southeast | 0.937 | 0.214 | 0.970 | 0.526 |

The yields showing considerable differences in the two investigated periods by region are shown in Figure 3. The lines show the linear trends; the strips indicate the 95% confidence intervals representing the correctness of fitting. The narrower the interval, the more precise the estimation is, while the intervals are wider in the case of a larger fluctuation of the basic data. Between 1970 and 1989, the average yield was increasing gradually, while after the sharp decline in the 1990s, the increase was moderate. Both periods can be characterized as experiencing large fluctuations. The results of the linear trend analysis of the yield changes by region are shown in Table 3.

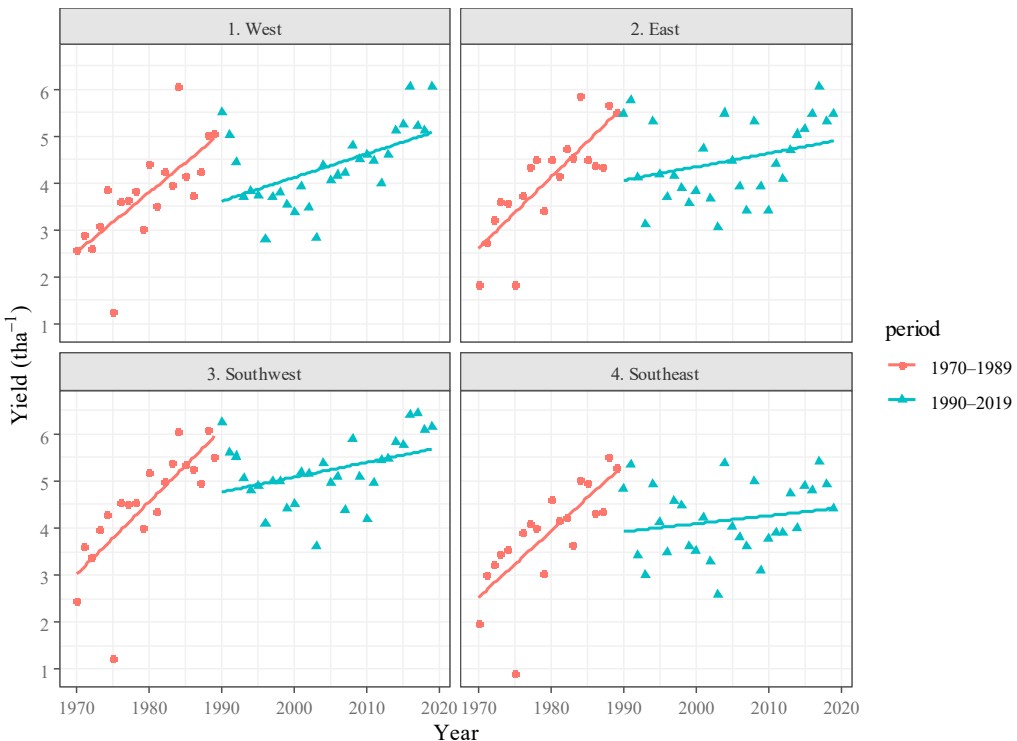

**Figure 3.** Linear trends in the average yields of winter wheat in the four investigated regions, split into two periods (1970–1989 and 1990–2019).

**Table 3.** Results of the annual linear trend analysis of winter wheat yield changes in the two investigated time periods by regions. R-squared stands for the coefficient of determination.

| Region | Estimate (t ha$^{-1}$ year$^{-1}$) | Residual Standard Error | R-Squared |
|---|---|---|---|
| **Linear Trend Analysis (1970–1989)** | | | |
| 1. West | 0.127 | 0.741 | 0.519 |
| 2. East | 0.151 | 0.651 | 0.665 |
| 3. Southwest | 0.154 | 0.774 | 0.594 |
| 4. Southeast | 0.141 | 0.740 | 0.573 |
| **Linear Trend Analysis (1990–2019)** | | | |
| Region | Estimate (t ha$^{-1}$ year$^{-1}$) | Residual Standard Error | R-Squared |
| 1. West | 0.050 | 0.712 | 0.285 |
| 2. East | n.s. | 0.831 | n.s. |
| 3. Southwest | 0.031 | 0.662 | 0.123 |
| 4. Southeast | n.s. | 0.758 | n.s. |

The estimates represent the slopes of the linear trends meaning the annual increase of the average yields. In the period of 1970–1989, the smallest increase was in the West region, 127 kg ha$^{-1}$ annually, which means approximately 5% per year, while a 141 kg ha$^{-1}$ (7%) annual increase was calculated for the Southeast region. In the East and Southwest regions, the annual increase was estimated to be the highest (151 and 154 kg ha$^{-1}$, representing 8% and 6% gain per year, respectively). In the second period, the average annual increase was much lower. At the same time, the error of trend fitting doubled due to the high fluctuation of the yield data. For the East and Southeast regions, the linear trends were not significant; the slopes were found zero. In the two Western regions, the correlation is significant, but the annual increase is slim, only 31–50 kg ha$^{-1}$. In a previous study, for three sub-regions

of Hungary out of seven, a linear decrease of 0.04 t ha$^{-1}$ year$^{-1}$ and no statistical trends for the rest were reported based on a simulation for the period of 1976–2005 [4].

Figure 4 shows the average yields of winter wheat and the residual standard errors (RSEs) of the linear trends for the two investigated periods and four regions.

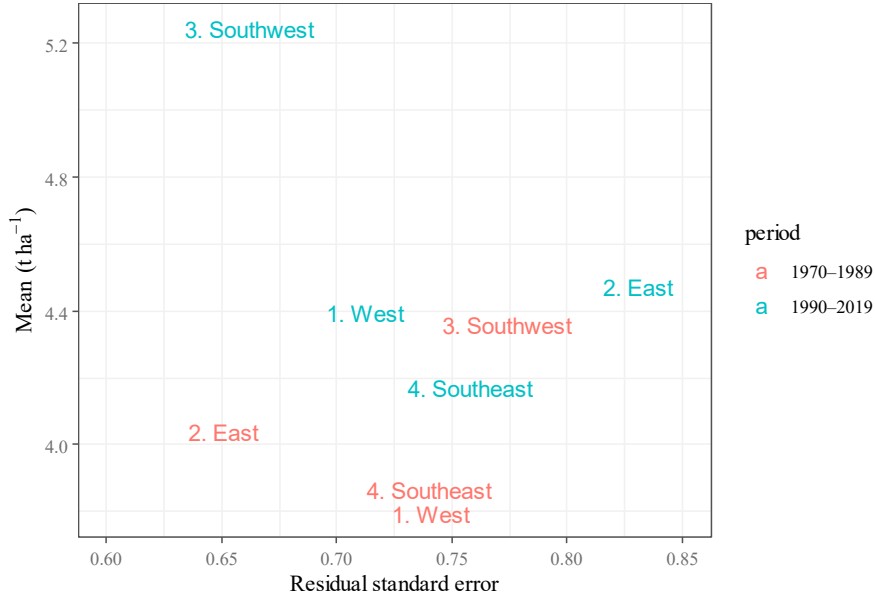

**Figure 4.** Variance mean plot of winter wheat yields for the investigated regions and periods.

The RSE values are practically the same for both periods (0.65–0.83 t ha$^{-1}$), although the average yields and the differences among the regions are significantly higher for the second period. Similarly, increases in the RSE were reported for winter wheat grain yields for the periods of 1961–1998 and 1999–2017 in France, 1961–1999 and 2000–2017 in Germany, and 1961–1996 and 1997–2017 in the United Kingdom [56]. In the case of substantial inter-annual yield variation, local-scale studies are needed to evaluate the uncertainties in yield predictions [25].

### 3.2. Long-Term Climatic Data in the Vegetation Period of Winter Wheat in Hungary (1970–2019)

The mean annual temperature and the GDD in the vegetation period of the winter wheat showed an increasing tendency in all four regions between 1970 and 2019 (Figures 5 and 6). The annual changes in temperature in the vegetation period in the West, Southwest, East, and Southeast were calculated to be 0.044 °C year$^{-1}$, 0.037 °C year$^{-1}$, 0.037 °C year$^{-1}$, and 0.035 °C year$^{-1}$, respectively. The 50-year average of GDD is the lowest in the West region (925.8 °C), while in the other three regions its values are similar in the range of 1048–1096 °C. The annual changes in GDD in the vegetation period in the West, Southwest, East, and Southeast were calculated to be 6.99 °C year$^{-1}$, 6.13 °C year$^{-1}$, 6.05 °C year$^{-1}$, and 5.76 °C year$^{-1}$, respectively.

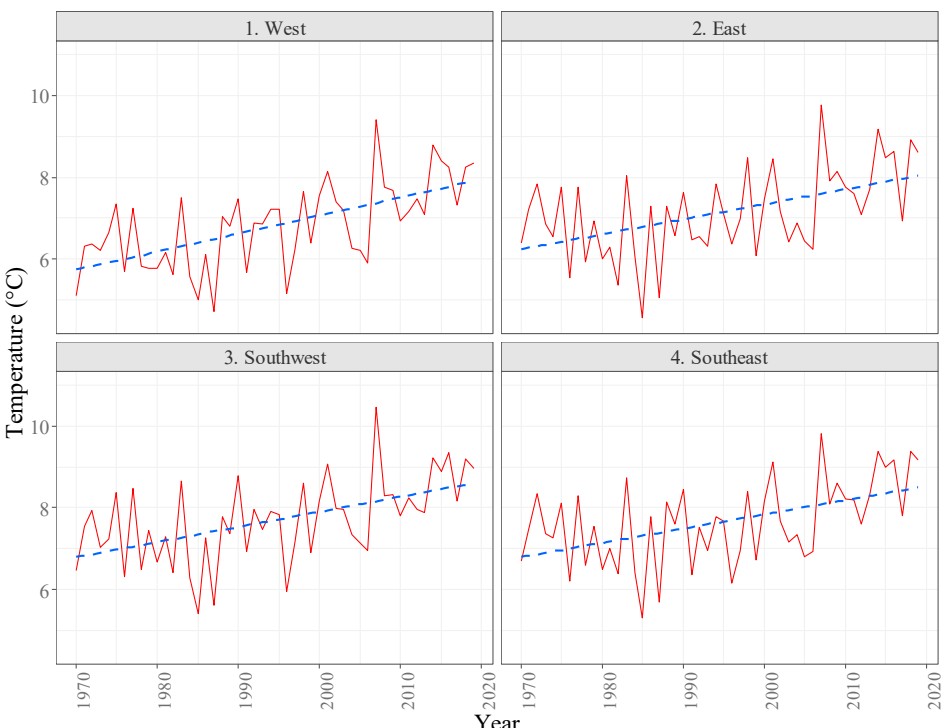

**Figure 5.** Mean air temperatures during the vegetation period of winter wheat (November–June) in the four investigated regions (1970–2019).

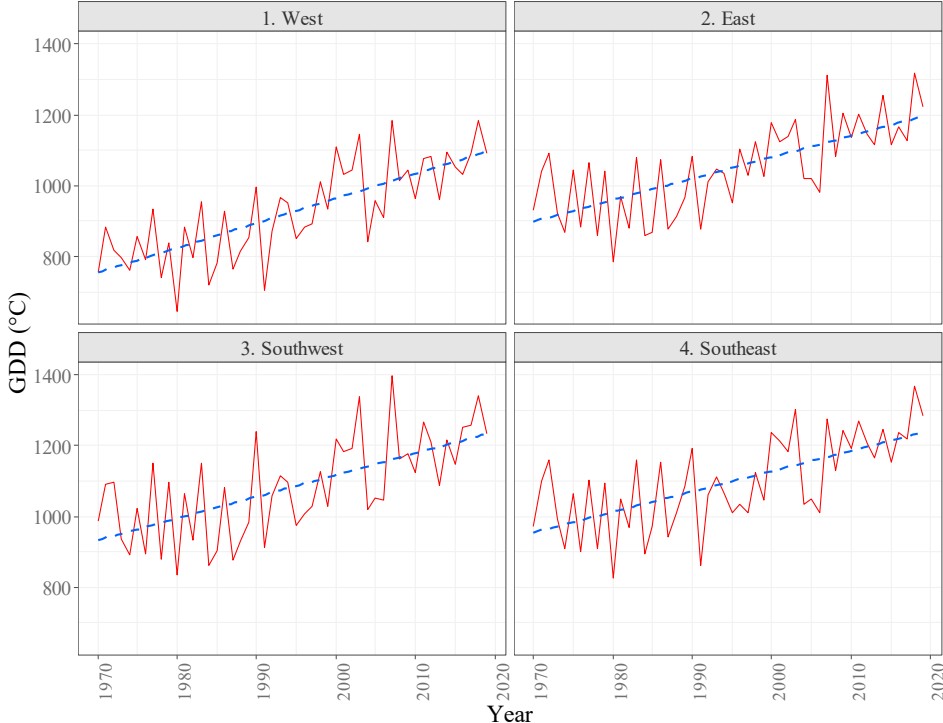

**Figure 6.** Winter wheat growing degree days (GDD), 1970–2019.

The rainfall in the vegetation period of winter wheat did not show considerable change in the investigated 50 years (Figure 7). The virtual decrease in the East region is due to the extremely high amount of annual rainfall in 1970 (953.2 mm).

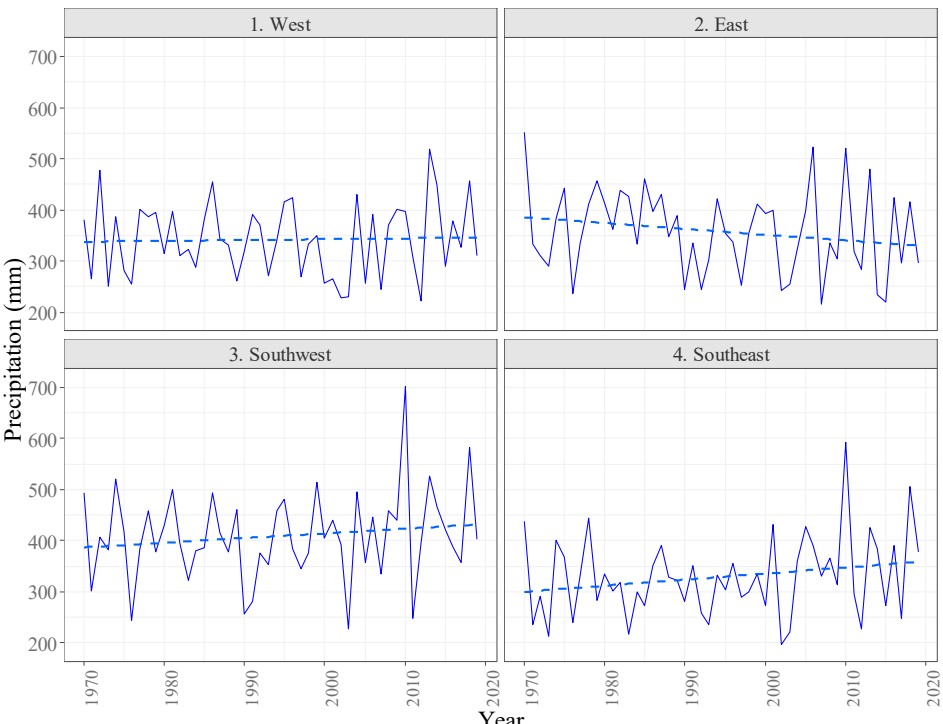

**Figure 7.** The amount of rainfall during the vegetation period of winter wheat (November–June) in the four investigated regions (1970–2019).

Figure 8 shows the biplot visualizing the relationship between the long-term mean air temperature and the average amount of precipitation during the vegetation period by region. The intersection of the lines represents the averages in terms of temperature and rainfall. The upper right quadrant represents the desirable circumstances during the vegetation period, with sufficient heat and water supply.

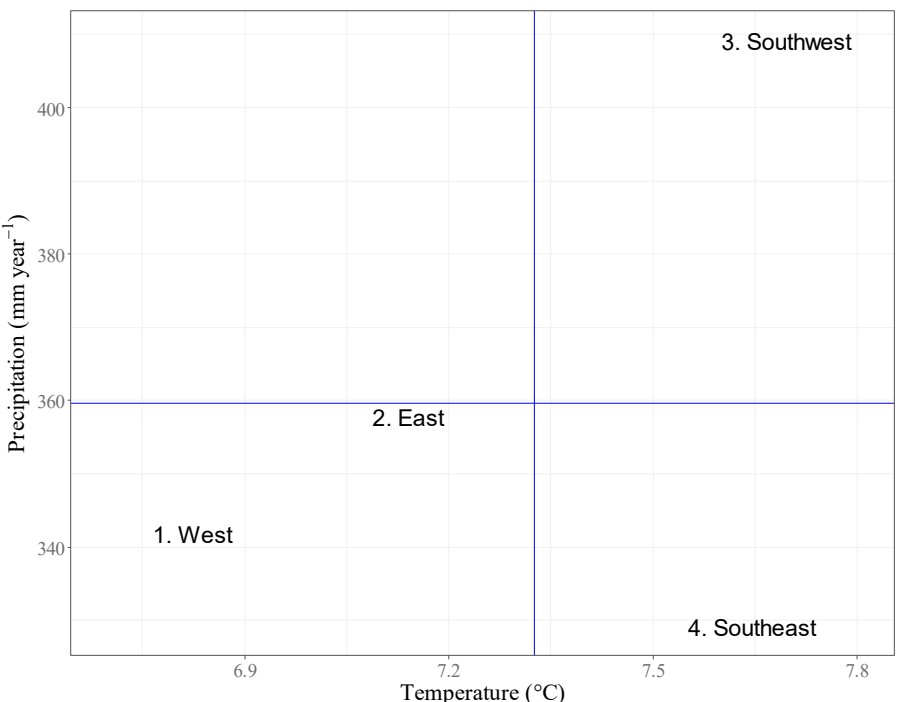

**Figure 8.** Biplot of the seasonal (*calculated for the vegetation period of winter wheat* (*November–June*)) mean air temperature and average amount of precipitation in the four investigated regions (1970–2019).

The Southwest region can be considered as the most suitable area for winter wheat production in Hungary. In the Southeast region, a higher temperature is accompanied by a low amount of precipitation, while both temperature and precipitation seem to be the limiting factors in the East and West regions. This is, however, not reflected by the mean yields by regions. Lower temperature with lower precipitation and higher T with higher precipitation represent the regions with higher variation in mean yields. In the West and Southwest, the precipitation per unit temperature is higher compared to the East and Southeast regions.

The mean air temperature and the average amount of precipitation do not provide full information on the CWB, which gives information about the sufficiency of the water supply for the undisturbed growth of a crop as an ecological factor. The seasonal (November–June) $ET_0$ values of the investigated regions determined for the years 1970–2019 are shown in Figure 9.

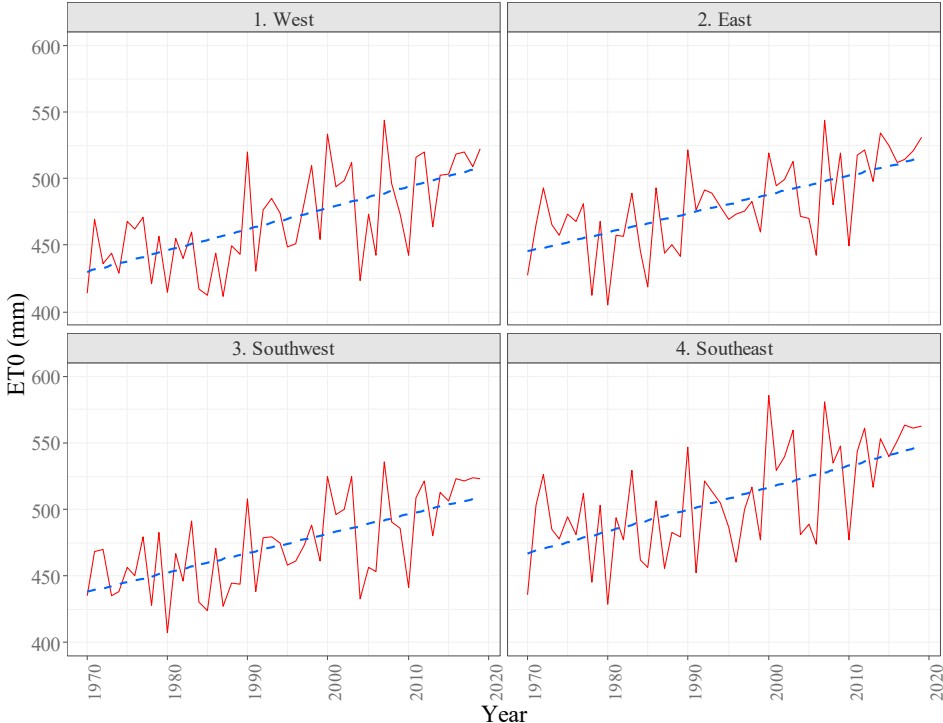

**Figure 9.** Seasonal values of reference evapotranspiration ($ET_0$) in the four investigated regions (1970–2019).

$ET_0$ showed a slightly increasing tendency in each region. The annual changes in the vegetation period in the West, Southwest, East, and Southeast were calculated at 1.60 mm year$^{-1}$, 1.46 mm year$^{-1}$, 1.43 mm year$^{-1}$, and 1.64 mm year$^{-1}$, respectively. As the global radiation did not change considerably, the increase is due to the increase in the temperature that also determines evapotranspiration and the decrease in the outgoing longwave radiation due to the increasing greenhouse effect. In a study conducted for Europe, a slight increase in the effective global radiation from sowing to anthesis at the northern locations and declines mostly at the southern sites were projected by 2060 [32]. According to another study, a small decline in global radiation during winter seems to decrease yield slightly [30]. For the vegetation period of winter wheat, the increase of $ET_0$ is lower than for the whole year, which makes winter wheat production more favorable than summer crops. Considering the global radiation too, yield variation between sites, for example, in Great Britain, was found to follow that of seasonal evapotranspiration [25]. For an area with a continental climate, they proposed the increased probability of the

adverse effect resulting from dry season from sowing to maturity, heat stress at anthesis, and adverse conditions during sowing.

The tendency in the trends of both the precipitation and the $ET_0$ was found different; hence CWBs were different for all four regions (Figure 10). The seasonal CWBs did not change considerably, although negative CWB, i.e., water shortage, was characteristic for all the regions. The annual change in the vegetation period was significant only in the East region, at $-2.56$ mm year$^{-1}$.

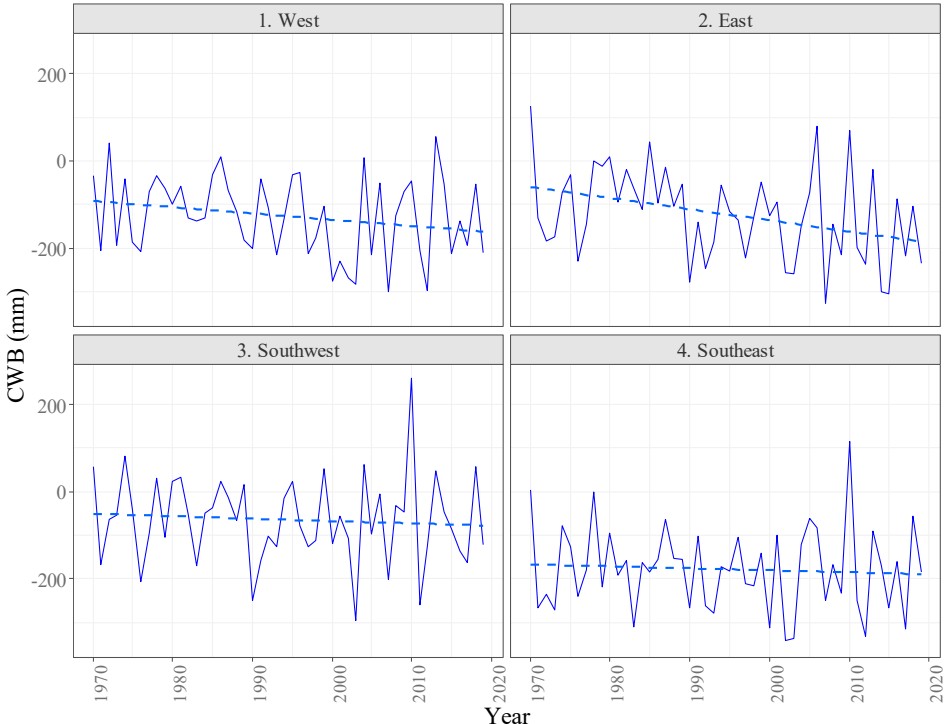

**Figure 10.** Seasonal (*calculated for the vegetation period of winter wheat* (*November–June*)) values of critical water balance (CWB) in the four investigated regions.

The biplot of the seasonal mean air temperature and the average amount of precipitation (Figure 8) seems to correspond to the variance mean plot of winter wheat yields by regions (Figure 4) showing that the vulnerability of production is the highest in the East and the lowest in the Southwest region. The Southwest is the wettest region and has the lowest CWB.

### 3.3. Critical Weather Parameters Based on the Correlation of Climatic and Yield Data

After detrending the linear yield data series, the deviation from the trend was taken as the dependent variable making the time series stationary for the analysis of the relationship between the weather conditions and the yield (Figure 11). Weather data of the vegetation periods were aggregated into weekly scales and detrended to find the periods critical for winter wheat growing. There is a break in the time series at 1989; therefore, only the 30 years after the change of the political and economic system in Hungary were analyzed. These three decades were considered homogenous. The yield increase, due to the rapid technological development between 1970 and 1989, impedes the ability to reveal the effect of the weather.

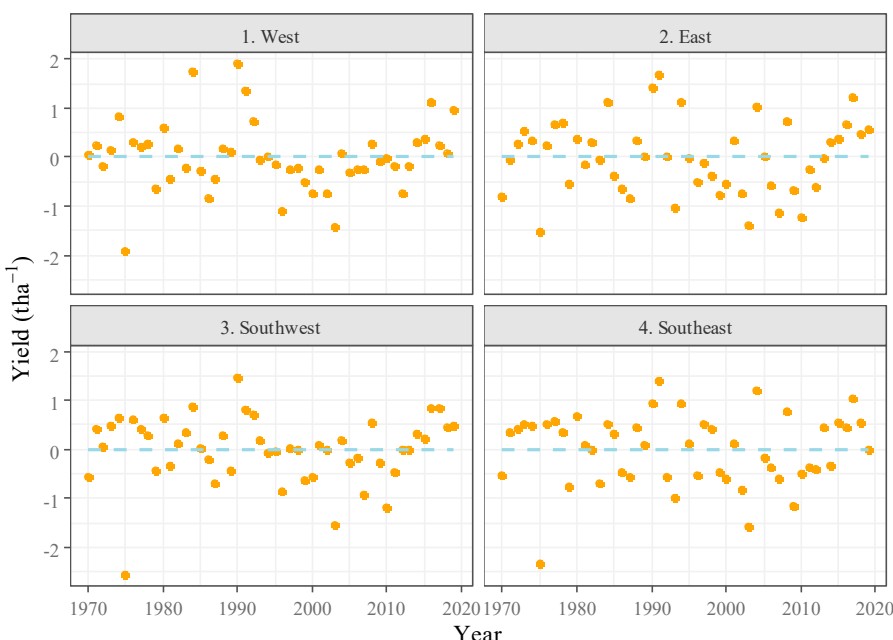

**Figure 11.** Detrended yields of winter wheat (1970–2019).

By multiple regression analysis, neither individually nor in any combinations, were the investigated weather parameters found to explain the change in yields and production vulnerability. There was no significant correlation between the detrended yields and any of detrended weather data. In some cases (weeks and regions), the *r*-values were around 0.3, but no tendencies could be proved.

In general, the difference among grain yield projection models in assumed temperature responses was proved to be the largest source of the uncertainty in simulated wheat yields [57]. In the case of the annual fluctuation of maize yield in Hungary, heat stress was a critical factor [41]. A statistical relationship was found between the heat stress and the winter wheat yield variance, too, but only for the period 1989–2019.

When yield is almost set, extreme heat stress can cause reduction in grain size and yield. It was demonstrated in a study that, despite the lower summer precipitation, as is projected for the 2050s in Europe, relative yield losses from drought can be expected to be lower as a result of the earlier winter wheat maturity, compared to the adverse effect of heat stress around flowering [33].

The critical weeks are summarized in Table 4. The *r*-values are valid for the extremely critical weeks. The *p*-values indicate the calculated significance levels, while $b_1$ is the steepness of the correlation between the most critical detrended heat units and yields. Each heat stress unit increase results in yield depression of winter wheat with the values of $b_1$ expressed in t ha$^{-1}$. The low value for the Southeast region results from that the sum of two late weeks with high heat stress units was calculated in the regression analysis. Variability in yield was linked to high and significant *r* for heat stress. The grain milk stage taking place on the 23rd week was found to be the most sensitive phenological period in all four regions. Variation in heat stress was higher in the cases of the East and Southeast regions, compared to the West and Southwest.

**Table 4.** The critical weeks of heat stress ($r \geq |0.33|$) for the investigated regions in the period of 1990–2019. *r*-value is the Pearson correlation coefficient, *p*-value is the probability of error type I, and $b_1$ is the yield change per hectare by heat stress unit. The *r*-values, *p*-values, and $b_1$ for the extremely critical weeks are indicated in bold.

| Region | Week | *r*-Value | *p*-Value | $b_1$ (t ha$^{-1}$ °C$^{-1}$) |
|---|---|---|---|---|
| 1. West | 19, 20, **23**, 25 | −0.516 | 0.005 | −0.257 |
| 2. East | **21**, 23 | −0.617 | 0.001 | −0.511 |
| 3. Southwest | 18, **19**, 20, 23, 24 | −0.684 | 0.003 | −0.364 |
| 4. Southeast | 19, 21, **23**, **24** | −0.612 | <0.001 | −0.161 |

In a study under controlled experimental conditions, winter wheat was reported to have two most sensitive periods to short episodes (2–5 days) of high-temperature stress, one between 8 and 6 days before anthesis and the other between 2 and 0 days before anthesis [15]. According to another study, exposure to extreme heat stress (+36/30 °C day/night temperature) at anthesis resulted in a grain yield loss of 81% [58]. It must be noted that winter wheat cultivars respond differently to heat stress related to both the timing and the duration. For example, testing 101 cultivars in a phytotron, the duration of heat stress explained 51.6% of grain yield phenotypic variance [59]. In general, heat stress can be expected to have the most considerable effect on the final grain yield in the period of heading to the grain milk stage in a continental region with a four-season climate.

### 3.4. Winter Wheat Yield Predictions for 2050 Based on a Robust Model

The expectable winter wheat yields can be predicted by means of a simple robust model. Our goal was to elaborate a model that can be applied for the whole country; therefore, the grain milk stage (23rd–24th) as the most sensitive period was selected to characterize the impact of heat stress (Figure 12). The model includes two components: (1) technological development and (2) the impact of heat stress.

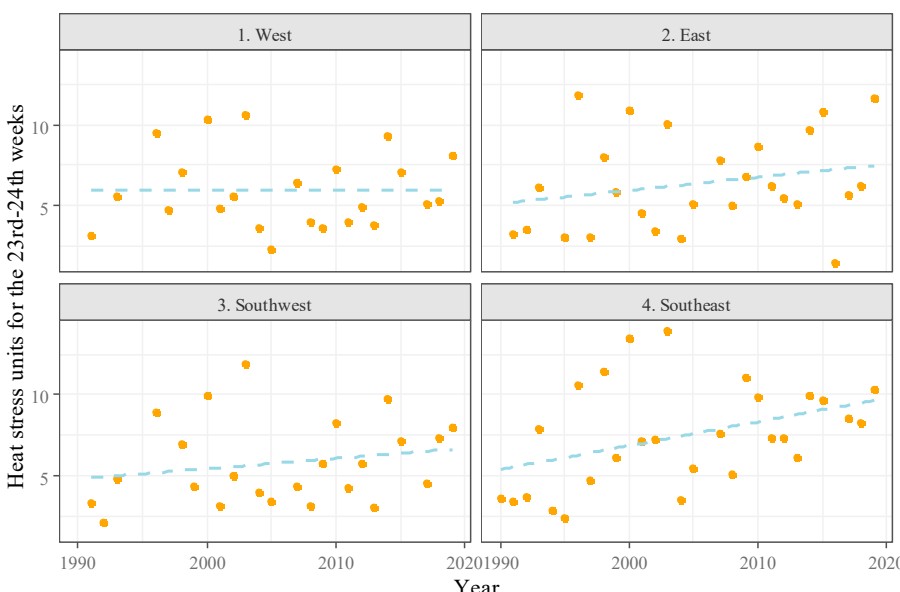

**Figure 12.** Heat stress units for the 23rd–24th weeks of the year in the four investigated regions (1990–2019).

A definite increasing linear trend was found in the East, Southwest, and Southeast regions, while the heat stress unit has not changed in the West region for the last 30 years. The highest steepness was characteristic for the Southeast region. For the predictions for the next 30 years, trends in the weather conditions in the past 30 years were considered.

In Table 5, the quantified impacts of the predicted technological development and heat stress on the yield of winter wheat are summarized by region, by 2050.

**Table 5.** Predicted impact of technological development and heat stress on the yield of winter wheat in the four investigated regions. $b_1$ is the yield change per hectare by heat stress unit for the critical 23rd–24th weeks. Predicted changes for 2050 are compared with the values of 2019, estimated by the trends.

| Region | Technological Development (t ha$^{-1}$ year$^{-1}$) | $b_1$ (t ha$^{-1}$ $^\circ$C$^{-1}$) | Heat Unit Increase ($^\circ$C year$^{-1}$) | Change by 2050 (t ha$^{-1}$) |
|---|---|---|---|---|
| 1. West | 0.050 | −0.097 | n.s. | 1.55 |
| 2. East | n.s. | −0.110 | 0.08 | −0.27 |
| 3. Southwest | 0.031 | −0.129 | 0.06 | 0.72 |
| 4. Southeast | n.s. | −0.161 | 0.15 | −0.75 |

Compared to the estimated yield for 2019, an increase in yield can be expected only in the western regions of Hungary (1.55 t ha$^{-1}$ in the West and 0.72 t ha$^{-1}$ in the Southwest region). For the two eastern regions, yield depression is predicted (0.27 t ha$^{-1}$ in the East and 0.75 t ha$^{-1}$ in the Southeast region). For the following 30 years, the predicted average annual yield fluctuation is 0.7–0.8 t ha$^{-1}$, which means that there will be years with yield fluctuation exceeding 1.5 t ha$^{-1}$ at a 95% probability. In a study based on a simulation model, Sirius, a comparable winter wheat yield decrease was predicted in the case that there is no change in $CO_2$ concentration by 2055 for Eastern Hungary [33]. When considering a 60% increase in $CO_2$ concentration, a 1 t ha$^{-1}$ increase was projected for the same location. It is advisable to carry out projections even on the level of the sub-regions of an agricultural area if findings aid decision support.

## 4. Conclusions

The mean winter wheat yield in Hungary increased in the past 50 years, although it experienced a sharp drop in 1989. High variation was found at country scale, and lower variation at regional scale. The growth intensity was lower after 1989. The variability in yield increase in the cases of the East and Southeast regions in the second period proposes increasing production risk, most intensively in the case of East, while it decreased in the cases of the West and Southwest regions. This suggests no link to the categories of climate zones defined by the IPCC.

Considering the trends in the long-term historical climatic data in the vegetation period of winter wheat at lower spatial scale, mean temperature and growing degree days showed a linear increase, with no considerable difference in the rates, by region. Precipitation in the vegetation period was found close to stable over the long term. In the West and Southwest, however, more precipitation per unit temperature was found in relation to the difference in the regional yields in the West and Southwest, compared to those in the East and Southeast regions. The increase in climatic water deficit resulted from the increase in the potential evapotranspiration. The highest variation in yield in the East region is in agreement with the most intensive negative change in CWB.

There was no drop in 1989 in the detrended yields representing the effect of weather without that of technological development. However, the variability was found to increase. Technological development was not found to affect the yield in the East and Southeast between 1990 and 2019 at all.

Winter wheat yield was not found to be sensitive to temperature, global degree days, precipitation, or CWB, but it was sensitive to heat stress at lower spatial scale and only during the period of 1989–2019. The highest sensitivity was found in the grain milk phenological period in the 23rd calendar week. The variation in heat stress was higher in the cases of the East and Southeast, compared to the West and Southwest regions, confirming its main role in yield variation.

A model including the effect of the trends in technological development and heat stress calculated for the last 30 years was developed to predict the change in yield by 2050. For the East and Southeast, a slight decrease was predicted, while for the West and Southwest regions, a more remarkable increase is expected.

In the case of Hungary, representing an agricultural continental region with four seasons, exposure to heat stress is expected to dominantly determine winter wheat yield in the future. Additionally, our results show that even contradictory trends can be expected within the same climatic zone or agricultural region by sub-region. This suggests that country-scale risk assessment and management planning should be based on sub-regional analysis. Furthermore, heat stress should be the main concern for researchers, plant breeders, and policy makers responsible for the future of winter wheat production in temperate climate zones. If historical climatic and yield data in high temporal resolution are available, our methodology can be applied for other regions as well as crops supporting successful crop production and yield improvement in the future.

**Author Contributions:** Conceptualization, L.H. and C.J.; methodology, L.H.; software, L.H.; validation, L.H.; formal analysis, L.H., E.K., and C.J.; investigation, L.H., J.Z., E.K. and C.J.; resources, L.H.; data curation, L.H.; writing—original draft preparation, E.K. and J.Z.; writing—review and editing, E.K. and J.Z.; visualization, L.H.; supervision, C.J.; project administration, E.K.; funding acquisition, C.J. All authors have read and agreed to the published version of the manuscript.

**Funding:** This research was funded by the National Research, Development and Innovation Fund of Hungary, 2020-4.1.1-TKP2020 funding scheme, grant number TKP2020-IKA-04, and the TKP2021-NKTA funding scheme, grant number TKP2021-NKTA-32.

**Institutional Review Board Statement:** Not applicable.

**Conflicts of Interest:** The authors declare no conflict of interest. The funders had no role in the design of the study; in the collection, analyses, or interpretation of data; in the writing of the manuscript or in the decision to publish the results.

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
