# Peer review of "Response of Winter Wheat (Triticum aestivum L.) Yield to the Increasing Weather Fluctuations in a Continental Region of Four-Season Climate"

_agronomy, doi:10.3390/agronomy12020314_

Round 1

Reviewer 1 Report

The current study entitled “Response of winter wheat (Triticum aestivum L.) yield to the increasing weather fluctuations in a continental region of four-season climate” is good. For a better understanding in-depth, it is a need for time to work on this topic. Furthermore, the achievement of potential benefits by using current technology is also dependent on the extensive research work for more exploration. Although the experiment is well organized, yet I suggest a rejection due to the following deficiencies.

Major Concerns

  • Systematic abstract is missing. Introduce the need for study in 1-2 lines.
  • Please give a clear-cut point problem source as a problem statement that is tackled in the current study.
  • Give logical reason for the selection of current strategy i.e., winter wheat (Triticum aestivum L.) selection on the basis of yield.
  • Quantitative data is also important to support your conclusion. Would you please provide some quantitative data in terms of percentage significant increase or decrease in the abstract?
  • Please provide a conclusive conclusion with is withdrawn through research in a single line. The statement “Our findings have the potential to be extendable to other countries with similar climate change trends” is general. Please conclude with a statement that shows a knowledge gap covered, potential beneficiaries and specific recommendations as well.
  • Give future prospective in a single line.
  • As per standard suggestions, please avoid using title words as keywords
  • Please follow the title in the introduction section, i.e., Response of winter wheat, then Yield problem, changing agro climate, knowledge gap, hypothesis and aims.
  • Also, provide a novelty statement at the end. What new things authors have done or correlated in this research compared to old ones?
  • Would you please give a single line about the knowledge gap which your research has covered along with the hypothesis statement?
  • Material and methods: It is ok.
  • In result, each figure and table must be self-explanatory. Please provide the details of the abbreviation at the end of each figure and table.
  • Please give a conclusive conclusion.
  • If the authors are not sure, then give future recommendations for more research and investigation.
  • Add the targeted beneficiary audience who will get benefits from this research.
  • Also, give clear-cut recommendations and future prospective regarding this research.

Reviewer 2 Report

The objectives of the study were to describe the trends in winter wheat grain yields in Hungary in the last 50 years; to describe the climatic conditions in the vegetation period by sub-region; to identify the critical weather variables in winter wheat production; and to give a prognosis for the regional winter wheat grain yields in 2050. The paper is well written, introduction provides sufficient background, methods are well described, results are clearly presented, discussed and supports the conclusion. The paper is interesting and could be used for production planning and decision making. My suggestions to the authors are to compare the obtained results with the studies conducted in similar agroecological conditions i.e. neighboring countries if possible, to format the manuscript according to the journal guidelines, and minor spell check is required.

Round 2

Reviewer 1 Report

Dear Authors,

I am satisfied with the corrections made. However, I request you and the editor to kindly improve the abstract part. 200 words are standard criteria but for scientific clarification, if the abstract part is a little bit increased it would not be a big concern. I have requested the editor to let you improve the abstract.

Regards

Author Response

Dear Reviewer,

Thank you for the opportunity to expand the text of the abstract of our manuscript. We followed your instructions and improved it. Please see the attachment.
